# Add-on pramipexole for anhedonic depression: study protocol for a randomised controlled trial and open-label follow-up in Lund, Sweden

Jesper Lindahl,[1,2] Marie Asp,[1,2] Darya Ståhl,[1] Johanna Tjernberg,[1,3] Moa Eklund,[1,2] Johannes Björkstrand,[4] Danielle van Westen,[5,6] Jimmy Jensen,[7] Kristoffer Månsson,[8] Åsa Tornberg,[9] Martina Svensson,[10] Tomas Deierborg,[10] Filip Ventorp,[1,2] Daniel Lindqvist [1,3]

FV and DL are joint senior authors.

For numbered affiliations see end of article.

**Correspondence to**
Dr Daniel Lindqvist;
daniel.lindqvist@med.lu.se

## ABSTRACT

**Introduction** Many depressed patients do not achieve remission with available treatments. Anhedonia is a common residual symptom associated with treatment resistance as well as low function and quality of life. There are currently no specific and effective treatments for anhedonia. Some trials have shown that dopamine agonist pramipexole is efficacious for treating depression, but more data is needed before it could become ready for clinical prime time. Given its mechanism of action, pramipexole might be a useful treatment for a depression subtype characterised by significant anhedonia and lack of motivation—symptoms associated with dopaminergic hypofunction. We recently showed, in an open-label pilot study, that add-on pramipexole is a feasible treatment for depression with significant anhedonia, and that pramipexole increases reward-related activity in the ventral striatum. We will now confirm or refute these preliminary results in a randomised controlled trial (RCT) and an open-label follow-up study.

**Methods and analysis** Eighty patients with major depression (bipolar or unipolar) or dysthymia and significant anhedonia according to the Snaith Hamilton Pleasure Scale (SHAPS) are randomised to either add-on pramipexole or placebo for 9 weeks. Change in anhedonia symptoms per the SHAPS is the primary outcome, and secondary outcomes include change in core depressive symptoms, apathy, sleep problems, life quality, anxiety and side effects. Accelerometers are used to assess treatment-associated changes in physical activity and sleep patterns. Blood and brain biomarkers are investigated as treatment predictors and to establish target engagement. After the RCT phase, patients continue with open-label treatment in a 6-month follow-up study aiming to assess long-term efficacy and tolerability of pramipexole.

**Ethics and dissemination** The study has been approved by the Swedish Ethical Review Authority and the Swedish Medical Products Agency. The study is externally monitored according to Good Clinical Practice guidelines. Results will be disseminated via conference presentations and peer-reviewed publications.

**Trial registration number** NCT05355337 and NCT05825235.

## STRENGTHS AND LIMITATIONS OF THIS STUDY

⇒ This study has a transdiagnostic design and includes a sample enriched for anhedonia symptoms.
⇒ A flexible dose titration strategy is used to reach a higher mean end dose of pramipexole.
⇒ Patients undergo functional MRI, blood sampling and lumbar punctures to investigate target engagement of pramipexole and potential pathophysiological mechanisms of anhedonic depression.
⇒ The open-label follow-up phase will inform about long-term efficacy and tolerability of pramipexole.
⇒ Comparability with previous reports may be limited since this is the first study to evaluate the efficacy of pramipexole in an anhedonic subgroup of depression.

## INTRODUCTION

Up to one-third of patients with depression do not achieve full remission, despite repeated treatment attempts with various medications.[1] Depression is a highly heterogeneous condition, both in terms of pathophysiology and symptom profile. The current diagnostic classification system does not sufficiently capture this heterogeneity which may obscure interpatient differences and lead to 'one-size-fits-all' treatment approaches, rather than targeting specific endophenotypes with mechanism-based treatments. Anhedonia—the inability to experience pleasure from, or the lack of motivation to carry out, usually enjoyable activities—has been suggested as a promising endophenotype of depression.[2]

Significant anhedonia is present in up to 40% of all depression and is associated with functional impairment and decreased quality of life.[3] It is also a risk factor for developing more severe depression, a more chronic illness course and treatment resistance to conventional antidepressants.[4] [5] Dopaminergic

neurotransmission in the mesolimbic pathway is involved in the ability to experience pleasure ('hedonia') and motivational drive.[6] Systemic low-grade inflammation has been suggested as an important upstream mechanism leading to alterations in dopaminergic neurotransmission and, in some cases, to the development of anhedonic depression.[7][8] In support of this mechanism, functional MRI (fMRI) studies of depressed patients show an association between systemic low-grade inflammation and anhedonia, mediated via decreased functional connectivity in the reward circuit.[9] Moreover, various types of immune challenges (to healthy individuals, hepatitis patients or animals) trigger anhedonia,[7] a blunted response to reward anticipation in the ventral striatum[10] and lower dopamine release and availability in the brain.[11][12] Dysfunction of the reward system is thought to be a hallmark of anhedonia pathophysiology.[13] fMRI studies have shown increased activation in the ventral striatum, and specifically in the nucleus accumbens (NAc), during reward anticipation, but this activation is generally smaller in patients with depression and anhedonia.[14] In the NAc, dopamine D3 receptors are expressed on neurons in high density.[15] This receptor is considered a potential treatment target for anhedonia, and an antidepressant effect of D3 receptor agonists has been demonstrated in animal studies.[16] Pramipexole is a dopamine receptor agonist with high affinity for the D3 receptor[17] that may also have anti-inflammatory properties.[18] It is an established treatment for Parkinson's disease (PD) motor symptoms and also has beneficial effects on depression and anhedonia in this patient group.[19] Previous studies testing the antidepressant efficacy of pramipexole in depression have been promising for both unipolar and bipolar depression,[20][21] but effect sizes have been too small for general clinical recommendations. It is possible that D3 agonists such as pramipexole are more efficacious in a subgroup of depression with more pronounced symptoms of anhedonia and biomarker alterations associated with dopaminergic hypofunction. For instance, a small-scale trial showed that D3 agonist piribedil had an antidepressant effect, with lower pretreatment levels of dopamine metabolite homovanillic acid (HVA) in cerebrospinal fluid (CSF) predicting treatment response.[22] In a case series of difficult-to-treat depression, Fawcett et al reported a clinically meaningful response in more than 75% of patients treated with high-dose pramipexole.[23] Based on their clinical experience and the dopamine agonistic effects of pramipexole, Fawcett et al suggested that this treatment may be particularly efficacious in a subtype of depression with a symptom profile of anhedonia and lack of motivation. This hypothesis is also consistent with the pathophysiology of motivational anhedonia, involving alterations in both dopaminergic neurotransmission and inflammation,[6] but no clinical studies have so far tested the efficacy of high-dose pramipexole in a patient population enriched for significant anhedonia symptoms.

Most previous treatment studies on depression have used relatively low doses of pramipexole while dose ranges are generally higher for PD. Several investigators have, however, suggested that a higher dose of pramipexole is needed to demonstrate an antidepressant effect of pramipexole.[23][24] Fawcett et al also suggested that dose titration should be tailored to the individual patient since optimal dosing varies according to several different factors such as age.[23]

Pramipexole is a promising candidate drug for anhedonic depression for several different reasons. Drug repurposing means identifying new use for an already approved drug outside the scope of the original indication.[25] This is a powerful and cost-efficient strategy for advancing therapeutic strategies which has been advocated for in psychiatry.[26][27] Our study is a clear example of drug repurposing which has several advantages over the development of a novel drug. First, the risk of failure is lower since (1) the safety profile of the repurposed drug has previously been established and (2) the time to implementation is generally shorter. Also, drug repurposing is considered cost-efficient if generic medications are available which is the case of pramipexole.

We have recently completed a small-scale open-label pilot study testing the feasibility of high-dose, add-on pramipexole in depressed patients with significant anhedonia (c.f. 'anhedonic depression'). We found that pramipexole was well tolerated and several patients in this difficult-to-treat sample, some of whom had previously received electroconvulsive therapy (ECT), had a meaningful response on anhedonia symptoms. Moreover, blood inflammatory markers decreased over the treatment course, and we observed a treatment-associated increase in reward-related activity in the ventral striatum.[28]

To the best of our knowledge, there are no previous RCTs testing the efficacy of add-on pramipexole for anhedonic depression. Moreover, no studies have tested long-term efficacy and safety profile of pramipexole in this patient group. These are the main aims of this study where patients with major depression (unipolar or bipolar) or dysthymia and significant anhedonia according to the SHAPS are randomised to receive either add-on pramipexole or placebo for 9 weeks. Thereafter, study participants are offered to continue treatment with add-on pramipexole in a 6 months open-label follow-up study. We also evaluate relevant blood and brain biomarkers as predictors of treatment response.

## METHODS AND ANALYSIS
### Overall study design
This is a double-blind trial in which a total of 80 patients are randomised to receive either add-on pramipexole or placebo for 9 weeks. Study participants undergo optional fMRI examinations and lumbar punctures at baseline and at the end of the RCT. After the 9 week RCT phase, study participants can enrol in a 6-month follow-up study designed to evaluate the long-term efficacy and tolerability of add-on pramipexole for anhedonic depression (see figure 1 for overview). Full inclusion and exclusion criteria are summarised in table 1.

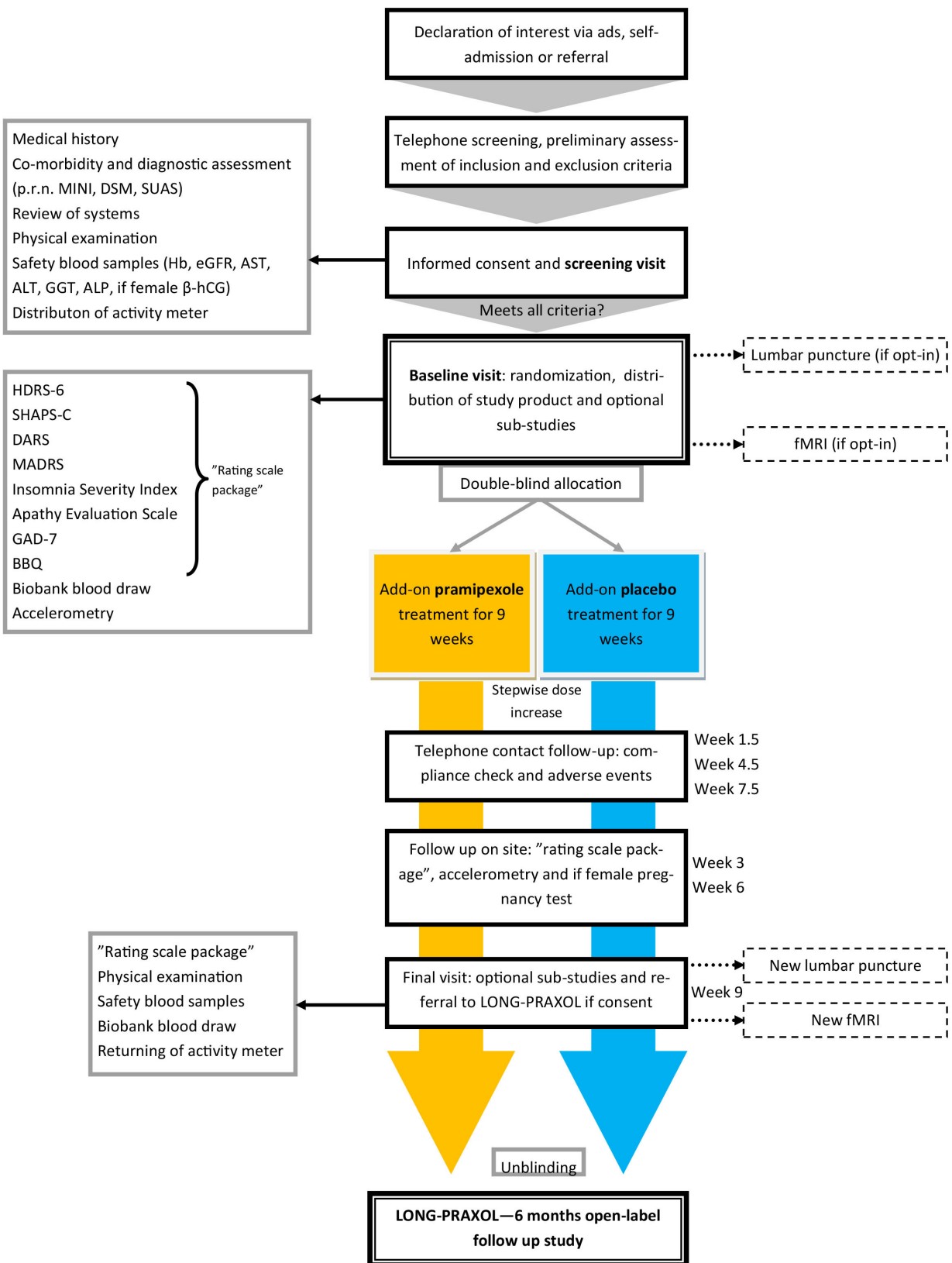

**Figure 1** Schematic flow chart describing the study protocol. MINI, Mini-International Neuropsychiatric Interview; DSM, Diagnostic and Statistical Manual of Mental Disorders; Hb, Hemoglobin; eGFR, estimated Glomerular Filtration Rate; AST, Aspartate aminotransferase; ALT, Alanine aminotransaminase; GGT, Gamma-glutamyl transferase; ALP, Alkaline phosphatase; BBQ, Brunnsviken Brief Quality of Life; fMRI, functional MRI; DARS, Dimensional Anhedonia Rating Scale; GAD-7, General Anxiety Disorder 7-item; HDRS-6, Hamilton Depression Rating 6-item Scale; MADRS, Montgomery-Åsberg Depression Rating Scale; SHAPS, Snaith Hamilton Pleasure Scale.

**Table 1** Inclusion and exclusion criteria

| | |
|---|---|
| Inclusion criteria | ▶ Age ≥18 years ≤75 years.<br>▶ Diagnosis of unipolar depressive episode or bipolar disorder in depressive phase or dysthymia.<br>▶ Clinically significant anhedonia symptoms: 3 or 4 points on ≥3 items of the SHAPS.[46]<br>▶ Ongoing treatment with at least one antidepressant or mood stabilising medication ≥4 weeks. Has tried an antidepressant at a therapeutic dose but not achieved remission (refractory stage 1 depression).[47]The research subject has given informed consent to participate in the study. |
| Exclusion criteria | ▶ Pregnancy, breast feeding or planned pregnancy (if female).<br>▶ High suicide risk according to the overall clinical assessment of the research physician.<br>▶ Ongoing substance abuse (within 6 months).<br>▶ Diagnosis of current psychotic disorder.<br>▶ Known diagnosis of Emotionally Unstable Personality Disorder.<br>▶ Ongoing treatment under the Compulsory Psychiatric Care Act.<br>▶ Medical history or strong clinical suspicion of impulse control disorder (including current binge-eating disorder) or a current ADHD diagnosis with hyperactivity.<br>▶ Diagnosis of intellectual disability, dementia or other circumstances that makes it difficult to understand the meaning of participating in the trial and provide informed consent.<br>▶ Diagnosis of renal failure (eGFR<50 mL/min/1.73 m$^2$) or severe cardiovascular disease (specifically symptomatic heart failure NYHA class II or greater).<br>▶ Recently started psychotherapy (within 6 weeks) or planning to start such treatment during participation in the trial.<br>▶ Ongoing ECT, ketamine or rTMS treatment, except maintenance ECT, ketamine or rTMS. (Maintenance treatment is defined as the use of ECT/ketamine/rTMS for a period exceeding 3 months after a series of ECT/ketamine/rTMS treatment in order to prevent the onset of a new episode).<br>▶ Other medical conditions or other concomitant drug treatment which, in the opinion of the investigators, may affect the evaluability of the trial or conditions that increase trial risk. For example, Parkinson's disease, hepatic insufficiency, ongoing cancer not in remission for more than one year, obesity surgery affecting the absorption of extended-release tablets.<br>▶ Ongoing treatment with drugs that affect plasma levels of pramipexole or have similar or antagonistic mechanism of action as pramipexole are not allowed. Ongoing treatment with neuroleptics is not allowed except for low-dose quetiapine[48] (≤150 mg/day) since it has very low binding to dopamine receptors at such low doses.<br>▶ Known or suspected allergy to any active substance or excipient in the medicinal product included in the trial.<br>▶ Participation in other treatment studies.<br>▶ Other reason, as assessed by the investigator, that prevents the research subject's participation, such as the risk that the research subject is unable to complete the trial (non-compliance). |

ADHD, Attention Deficit Hyperactivity Disorder; ECT, electroconvulsive therapy; NYHA, New York Heart Association; rTMS, repetitive Transcranial Magnetic Stimulation; SHAPS, Snaith Hamilton Pleasure Scale.

The study began recruiting participants in February 2023. Based on the current recruitment pace, we estimate that the last patient will complete the study during the second half of 2025.

### Primary and secondary outcome measures

Primary and secondary outcome measures are summarised in table 2. The primary outcome measure, SHAPS, comprises 14 items scored from 1 to 4, with a higher score reflecting more severe anhedonia.[29] Rating scales are administered, and adverse events assessed at weeks 3, 6 and 9 (end of study). Blood, CSF sampling and fMRI are done at baseline and at week nine. Objective assessments of sleep patterns and physical activity are conducted continuously throughout the 9 weeks of the RCT phase using accelerometry as further described below. During the 6 months open-label follow-up phase of the study, we collect the same outcome measures as in the RCT (once/month), except for accelerometry which is collected for 7 consecutive days after the 3-month and 5-month visits. Lumbar punctures and fMRI are not done during the 6-month follow-up phase.

### Trial drug and placebo

Pramipexole tablets are purchased from STADA Nordic and repackaged and labelled according to EU guidelines and good manufacturing practice (GMP). Placebo manufacturing is done according to GMP by Ardena. Placebo tablets are identical to pramipexole tablets in the strengths 0.26 mg base (=0.375 mg salt), 0.52 mg base (=0.75 mg salt), 1.05 mg base (=1.5 mg salt) and 2.1 mg base (=3.0 mg salt) pramipexole extended-release tablets from STADA Nordic. We instruct study participants to take the tablets once per day, at night-time.

All study personnel involved in patient assessments are blind with regard to pramipexole or placebo allocation. Two members of the study team, who are not involved in any patient assessments, are unblinded and place packets of placebo or active treatment of different concentrations in a box designated for each research subject. From

**Table 2** Primary and secondary outcome measures

|  | Description | Name of scale |
|---|---|---|
| Primary outcome measure | Anhedonia symptoms | SHAPS[49] |
| Secondary outcome measures | Core depression symptoms | HDRS-6[50] |
|  | Anhedonia symptoms (covering additional anhedonic domains compared with the SHAPS) | DARS[51] |
|  | General depressive symptoms | MADRS-S[52] |
|  | Sleep disturbances | ISI[53] |
|  | Apathy symptoms | AES[54] |
|  | Anxiety symptoms | GAD-7[55] |
|  | Quality of life | BBQ[56] |
|  | Accelerometry—no of steps per day, distribution of movement pattern over the day, sedentary behaviour, time spent in light, moderate and vigorous physical activity, sleep latency (time to fall asleep), sleep awakening (how often one wakes up during the night), wakefulness (time in minutes awake during one night), time in deep sleep, sleep efficiency (sleep time vs total time spent in bed) |  |
|  | No and severity of adverse events | N/A |
|  | fMRI with focus on the monetary incentive delay task[32] and task-based brain signal variability.[35] | N/A |
|  | Blood and CSF biomarkers | N/A |

AES, Apathy Evaluation Scale; BBQ, Brunnsviken Brief Quality of Life; CSF, cerebrospinal fluid; DARS, Dimensional Anhedonia Rating Scale; fMRI, functional MRI; GAD-7, General Anxiety Disorder 7-item Scale; HDRS-6, Hamilton Depression Rating 6-item Scale; ISI, Insomnia Severity Index; MADRS, Montgomery-Åsberg Depression Rating Scale; NA, not available; SHAPS, Snaith-Hamilton Pleasure Scale.

this box, blinded investigators can then retrieve drugs for administration to research subjects during the trial without the unblinded staff member's presence, as we have a flexible dosing schedule. Subjects are allocated to either pramipexole or placebo based on a pre-established randomisation list.

Based on previous studies including our own pilot data,[23 28 30] we hypothesise that a relatively higher dose is needed to be able to evaluate any beneficial effects of add-on pramipexole on anhedonia or any of the other outcome measures. Based on previous reports,[23] we also believe that dose titration should be tailored individually, therefore, we use a flexible dose titration schedule. Dose is increased on a weekly basis according to Swedish

**Table 3** Dose titration schedule in RCT

| Week | Step | Total dose (mg base) | Total dose (mg salt) |
|---|---|---|---|
| 1 | 1 | 0.26 | 0.375 |
| 2 | 2 | 0.52 | 0.75 |
| 3 | 3 | 1.05 | 1.5 |
| 4 | 4 | 1.57 | 2.25 |
| 5 | 5 | 2.1 | 3.0 |
| 6 | 6 | 2.62 | 3.75 |
| 7 | 7 | 3.15 | 4.5 |
| 8 |  | 3.15 | 4.5 |
| 9 |  | 3.15 | 4.5 |

RCT, randomised controlled trial.

guidelines for PD (see table 3). In case of intolerable adverse reactions, the dose will be lowered and a new attempt to increase the dose will be made after approximately 1 week. In case of significant improvement during the study (Clinical Global Impression-Severity≤2 points), we stop dose increase until the next scheduled evaluation.

### Biological sampling and biomarkers

Blood and CSF samples are collected at baseline and week 9, aliquoted, frozen at −80°C and stored in biobank. Subsequent analyses will include markers of inflammation, cellular health (including neurodegeneration), cellular stress and metabolism, growth factors and monoamine metabolism, as well as analyses of genetic variants relevant to these biological systems. Dopamine metabolite HVA is an example of a biomarker that will be analysed in CSF. Lower baseline levels of CSF-HVA predicted antidepressant response to D3 agonist piribedil in a small-scale clinical trial,[22] and our study will provide an opportunity to test this hypothesis in a larger sample. In addition to the well-established effects of pramipexole on dopamine transmission, preclinical data suggest that it may also have anti-inflammatory properties.[18 31] This is also consistent with our own pilot data showing a decrease in high-sensitivity C reactive protein over the course of pramipexole treatment.[28] Therefore, inflammatory markers, in blood and CSF, will be analysed before and after the intervention to investigate if low-grade inflammation is related to pramipexole treatment response.

Lumbar punctures are i performed before and after 9 weeks of study intervention. i CSF samples are aliquoted, frozen at −80°C and stored in a biobank. Patients who undergo lumbar punctures sign a separate informed consent and receive monetary compensation of 2500 SEK (~US$230) for two lumbar punctures (pretreatment and post-treatment).

## fMRI experimental designs

Data will be acquired using a 7T whole body MR scanner (Philips Achieva, Philips Medical Systems, Best, The Netherlands) with a 32-channel head coil. To examine the effect of treatment on neural activity during reward anticipation, we use a simplified version of the monetary incentive delay task (MID) focusing on the appetitive aspects of the task.[28 32] The task structure is based on previous studies using simplified versions of the incentive delay task[32] focusing solely on aversive stimulation.[33] In the current study, the task consists of two cues that signals either a high reward (€0.5) or a low reward (€0.01) presented to the participant. Whether the participant receives the reward or not is contingent on their response time to a subsequent target cue. The task is programmed to deliver rewards in 75%–80% of the trials where the delivery schedule is maintained by the that the target cue is adaptive to each individual's response times. To examine the effect of treatment on striatal neural activity, we use fMRI blood-oxygen-level-dependent (BOLD)-signal imaging during the reward-anticipation phase in our regions of interest, encompassing the striatum, midbrain and prefrontal and orbitofrontal cortices. The low-reward contrast is thus used as a control condition with the same trial structure as the high reward condition (controlling for non-specific activations related to visual stimulation), but since the reward is too low (approximately €0.01) little reward related activity is expected.

Neural signals notably vary across time. Moment-to-moment variability in the BOLD signal has been linked to cognition and behaviour.[34] Månsson et al reported good test–retest reliability for BOLD signal variability, and that it predicted anxiety disordered patients' response to psychological treatment.[35] Here, we will use a similar suppression-repetition task using visual stimulation (eg, emotional faces), and test both neural variability as a pretreatment predictor, and sensitive marker of psychiatric treatment response in depression.

Patients who undergo fMRI sign a separate informed consent and receive monetary compensation of 750 SEK+up to 500 SEK in 'winnings' during the MID (~US$110 in total).

## Physical activity and sleeping patterns

Physical activity and sleeping pattern are assessed using accelerometry (ActiGraph GT3X-BT; ActiGraph, Pensacola, Florida, USA). The accelerometer is placed at the wrist of the non-dominant hand and is only removed when showering or bathing. Accelerometer data will be analysed for sedentary behaviour; and light, moderate and vigorous physical activity levels.[36] Sleep patterns, as total sleep time, sleep efficiency, sleep latency and wake time after sleep onset will be analysed according to Cole-Kripke algorithm.[37]

## Follow-up

After completing the 9 weeks RCT, patients who have received trial drug and continue to have no exclusion criteria are offered to continue with pramipexole for 6 months in an open-label follow-up study. Patients who have received placebo and fulfil the eligibility criteria of the RCT, can also enrol in the follow-up study and will then be treated with open-label add-on pramipexole using the same dose titration schedule as in the RCT. The members of the study team working with the follow-up are unblinded and they do not work with patient assessment in the RCT.

## Statistical considerations

We will use linear mixed model with repeated measures (weeks 3, 6 and 9) as the main statistical method to test the clinical efficacy of pramipexole. The estimated effect size of pramipexole treatment is estimated to 0.27 as this has been reported in other studies of antidepressants.[38] This corresponds to a MADRS score difference of approximately 4 points, which is considered a significant clinical improvement[39] and corresponds to ≥4 p on the SHAPS-C. Estimation of correlation coefficient between repeated measures of MADRS scores is based on data from our previous pilot study (r=0.5). For a power calculation with strength 80% and α=0.05, it is estimated that 74 research subjects are needed. The drop-out rate is estimated to be around 5% (based on our pilot study), so we need to include 80 research subjects in the study who start treatment.

Paired sample t-test (or appropriate non-parametric method) will be used for analysis of difference of biological measures between baseline and endpoint. Repeated measure analysis of variances with time×group interaction effects will be used to examine whether change in biomarkers differs between those responding to treatment and those not responding to treatment. We will also test baseline differences in biomarkers between responders/non-responders and remitters/non-remitters.

Primary and secondary outcome measures related to the efficacy of pramipexole will be analysed in the intention-to-treat-population, that is, all randomised patients. In addition to analysing total scores as continuous variables, we will also do descriptive analyses of treatment response and remission status. For this purpose, we use established cut-offs. Response according to the SHAPS is defined as an improvement of ≥50% on total score (each item rated from 1 to 4), and remission as a SHAPS score of ≤3, based on an alternative scoring approach rating each item either 0 or 1.[40 41] Response on the MADRS is defined as ≥50% improvement and remission as a score ≤10. When half of the participants have completed the trial, an interim analysis of the SD on the primary outcome measure (total

SHAPS score) and calculation of the number of drop-outs is performed. A new power calculation can be made based on this information in order to have the possibility to adjust the number of research subjects (with a new application for modification to the Swedish Medical Products Agency and the Swedish Ethical Review Authority) if necessary.

### Patient and public involvement

Patients and/or public were not involved in the design of this protocol.

## ETHICS AND DISSEMINATION

The study protocol (version 3.2 dated 9 May 2023) has been approved by the Swedish Ethical Review Authority (reference number 2023-01927-02) and the Swedish Medical Products Agency (EudraCT 2022-001563-26)). The study is monitored according to Good Clinical Practice by Clinical Studies Sweden, Forum South. All participants receive oral and written information and thereafter sign an informed consent which is obtained by the research physician, prior to entering the study. The participant consent form in Swedish can be found in online supplemental material. The trial sponsor is Region Scania through the Psychiatric Clinic in Lund.

Treatment with pramipexole may cause dose-dependent adverse reactions such as nausea, vomiting, orthostatic hypotension, headache or dizziness. In our pilot study, which included 12 patients with 'anhedonic depression', headache, nausea and sleep disturbance were the most common adverse reactions. There are several previous reports, derived mainly from PD populations, of rare but more serious adverse events such as motor disturbances, hallucinations, delusions, confusion, mania and impaired impulse control.[42] To reduce the risk for these types of adverse events, we actively screen patients for psychotic illness, addiction and impulse control disorders, and we follow patients closely over the trial period, focusing on these types of rare adverse reactions with relevant assessment instruments. We include patients with bipolar disorder in depressive phase in our trial. A review of the literature has shown that the proportion of patients who experience a transition from depression to a manic state when treated with pramipexole is comparable to (and in some cases smaller than) those who receive placebo.[43] Unlike other dopamine receptor agonists, pramipexole can induce somnolence (especially at higher doses) and, in rare cases, patients have fallen asleep without warning.[42] Therefore, patients suffering from somnolence are instructed to refrain from driving and operating heavy machinery until this adverse reaction has ceased. Pramipexole does not have an approved indication for the treatment of anhedonia symptoms and will be administered in this trial mainly according to the Summary of Product Characteristics for the treatment of PD (maximum dose 3.15 mg base/day). This has also been done in previous clinical studies on depressed patients.[30 44 45] Some of these studies also treated with pramipexole doses above 3.15 mg base.[23 30] As this is likely to increase the risk of neuropsychiatric adverse reactions[42] and is not an approved dose range in Sweden, this has not been done in the present trial. Adverse reactions are systematically monitored and recorded within the trial.

The risk of severe adverse reactions is considered low within the dose range used in this study, as confirmed by several previous clinical studies with pramipexole in depressed patients[23] as well as our own pilot data. Thus, we believe that the potential benefits of identifying an efficacious treatment for anhedonic depression outweigh the risks for the included research subjects.

**Author affiliations**
[1]Unit for Biological and Precision Psychiatry, Department of Clinical Sciences Lund, Lund University, Lund, Sweden
[2]Office for Psychiatry and Habilitation, Psychiatric Clinic Lund, Region Skåne, Lund, Sweden
[3]Office for Psychiatry and Habilitation, Psychiatry Research Skåne, Region Skåne, Lund, Sweden
[4]Department of Psychology, Lund University, Lund, Sweden
[5]Diagnostic Radiology, Department of Clinical Sciences Lund, Lund University, Lund, Sweden
[6]Image and Function, Skåne University Hospital, Lund, Sweden
[7]Department of Psychology, Kristianstad University, Kristianstad, Sweden
[8]Center for Psychiatry Research, Department of Clinical Neuroscience, Karolinska Institutet, Stockholm, Sweden
[9]Department of Health Sciences, Lund University, Lund, Sweden
[10]Experimental Neuroinflammation Laboratory, Department of Experimental Medical Science, Lund University, Lund, Sweden

**Contributors** DL is involved in data collection, project coordination, secured funding, conceptualised the study and cowrote the original protocol. FV is involved in data collection, project coordination, conceptualised the study and cowrote the original protocol. JL is involved in data collection, project coordination and provided substantial input on the protocol. MA is involved in data collection and project coordination and provided significant input on the protocol. DS is involved in data collection and project coordination and provided significant input on the protocol. JT is involved in data collection and project coordination and provided significant input on the protocol. ME is involved in data collection and project coordination and provided significant input on the protocol. DvW, JJ, KM and JB conceptualised the fMRI experiments and cowrote this part of the protocol and also provided significant input on other parts of the protocol. ÅT, MS and TD contributed to conceptualisation for ecological momentary assessments and provided significant input on the protocol. All authors approved the final version of the manuscript.

**Funding** The study has received funding from the Swedish Research Council (grant number 2020-01428), Swedish governmental funding of clinical research (ALF) (award/grant number is not applicable), grants from the province of Scania (award/grant number is not applicable), the Crafoord Foundation (grant number 20220522), the Brain Foundation (FO2022-0050), Ellen and Henrik Sjöbring Foundation (Award/Grant number is not applicable), Söderström–Königska Foundation (grant number SLS-969692), Region Kronoberg (award/grant number is not applicable), Bror Gadelius Foundation (award/grant number is not applicable) and the Olle Engkvist Foundation (grant number 214-0363).

**Competing interests** MA has received speaking honorarium from H. Lundbeck AB. DL has received speaking honorarium from H. Lundbeck AB and Janssen-Cilag AB

**Patient and public involvement** Patients and/or the public were not involved in the design, or conduct, or reporting, or dissemination plans of this research.

**Patient consent for publication** Not applicable.

**Provenance and peer review** Not commissioned; externally peer reviewed.

**ORCID iD**
Daniel Lindqvist http://orcid.org/0000-0002-3472-327X

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
