## [Reviewer comments · BMJ Open]

ARTICLE DETAILS

TITLE (PROVISIONAL)	Add-on pramipexole for anhedonic depression - study protocol for a randomized controlled trial and open label follow-up in Lund, Sweden.
AUTHORS	Lindahl, Jesper; Asp, Marie; Ståhl, Darya; Tjernberg, Johanna; Eklund, Moa; Björkstrand, Johannes; van Westen, Danielle; Jensen, Jimmy; Månsson, Kristoffer; Tornberg, Åsa; Svensson, Martina; Deierborg, Tomas; Ventorp, Filip; Lindqvist, Daniel

VERSION 1 – REVIEW

REVIEWER	Sundberg, Isak Uppsala University
REVIEW RETURNED	13-Sep-2023

GENERAL COMMENTS	This is an ambitious study, well designed to answer important research questions. There is a relative lack of efficacious treatments for depression, and there is a need for more personalized treatment options based on clinical subtypes. Therefore a primary outcome of anhedonia is interesting. It is positive that the study protocol includes measures of target engagement. While the manuscript is generally well written, I have some remarks: -I cannot see that the authors have included the dates of the study in the manuscript, as stipulated by BMJ Open.-It is stated that descriptive analyses will be performed of treatment response or remission status (using established cut-offs). These cut offs should be defined to avoid ambiguity; for example if response will be defined as a reduction of at least 50% in total symptoms.-Regarding the language, there may be room for some improvement. Preferrably, the same tense should be used in each section for clarity. There seems to be some minor linguistic mistakes, for example: page 17 row 49 to fulfill page 18 row 15 estimated estimated-There seems to be another similar trial planned, albeit not with anhedonia as primary outcome:\nhttps://mentalhealth.bmj.com/content/25/2/77.long Perhaps data could be pooled and analyzed later post hoc, as the other research group also will include SHAPS, albeit as secondary outcome?-There is a recent systematic review and meta-analysis on observational studies of add-on pramipexle in treatment resistant depression, that the authors may want to add to the reference list:
--

	https://pubmed.ncbi.nlm.nih.gov/37109571/ I wish the authors the best of luck with this important study!
--	---

REVIEWER	Cavaleri, Daniele University of Milan-Bicocca, Department of Medicine and Surgery
REVIEW RETURNED	13-Sep-2023

GENERAL COMMENTS	This submission reports a protocol for a randomized controlled trial and open label follow-up on add-on pramipexole repurposed as an augmentation strategy for anhedonic depression. While I cannot spot any major methodological issue, some further considerations should be added to the manuscript:  1. For readers not acquainted with the concept of drug repurposing and its application in psychiatric conditions, I suggest that the Authors provide some background information in their Introduction section. Please refer to Fava's eminent editorial published in World Psychiatry [https://doi.org/10.1002/wps.20481]. 2. Perspectives in research on drug repurposing in psychiatry should be briefly discussed in view of recent, relevant evidence in the field. In this regard, the Authors should refer to a recent umbrella review on the topic by Bartoli et al. [2021, https://doi.org/10.1016/j.jpsychires.2021.09.018]. The Authors should consider that for pramipexole in depression (as for many other repurposed drugs) currently available evidence is limited – with most meta-analyses being underpowered and only few of them having significant results – and as of today it does not allow making reliable recommendations for the use of these strategies in clinical practice.
---

REVIEWER	Hieronimus, Fredrik University of Gothenburg, Institute of Neuroscience and Pharmacology
	I have previously received occasional speaker's fees from Servier, H Lundbeck and Janssen Pharmaceuticals. The only one that is from less than 3 years ago was from Janssen Pharmaceuticals.
REVIEW RETURNED	18-Sep-2023

GENERAL COMMENTS	An interesting and highly clinically relevant study which appears to me well designed. Will be looking forward to hearing of the results.
---

VERSION 1 – AUTHOR RESPONSE

Reviewer: 1
Dr. Isak Sundberg, Uppsala University

Comments to the Author:

This is an ambitious study, well designed to answer important research questions. There is a relative lack of efficacious treatments for depression, and there is a need for more personalized treatment options based on clinical subtypes. Therefore a primary outcome of anhedonia is interesting. It is positive that the study protocol includes measures of target engagement.

While the manuscript is generally well written, I have some remarks:

-I cannot see that the authors have included the dates of the study in the manuscript, as stipulated by BMJ Open.

Author response: Thanks for this comment. Dates for the start and completion of the study have been added in the suggested section.

-It is stated that descriptive analyses will be performed of treatment response or remission status (using established cut-offs). These cut offs should be defined to avoid ambiguity; for example if response will be defined as a reduction of at least 50% in total symptoms.

Author response: Thanks for this comment. The established cut offs have been added for clarification.

-Regarding the language, there may be room for some improvement. Preferrably, the same tense should be used in each section for clarity. There seems to be some minor linguistic mistakes, for example:

page 17 row 49 to fulfill

page 18 row 15 estimated estimated

Author response: We have proof-read the manuscript again to correct these issues.

-There seems to be another similar trial planned, albeit not with anhedonia as primary outcome:

<https://mentalhealth.bmj.com/content/25/2/77.long>

Perhaps data could be pooled and analyzed later post hoc, as the other research group also will include SHAPS, albeit as secondary outcome?

Author response: Thank you for this comment. This is a very good suggestion that we will take into consideration after completing the study.

-There is a recent systematic review and meta-analysis on observational studies of add-on pramipexle in treatment resistant depression, that the authors may want to add to the reference list:

<https://pubmed.ncbi.nlm.nih.gov/37109571/>

Author response: Thanks for this comment. This reference has been added.

Reviewer: 2

Dr. Daniele Cavaleri, University of Milan-Bicocca

Comments to the Author:

This submission reports a protocol for a randomized controlled trial and open label follow-up on add-on pramipexole repurposed as an augmentation strategy for anhedonic depression.

While I cannot spot any major methodological issue, some further considerations should be added to the manuscript:

1. For readers not acquainted with the concept of drug repurposing and its application in psychiatric conditions, I suggest that the Authors provide some background information in their Introduction section. Please refer to Fava's eminent editorial published in World Psychiatry [<https://doi.org/10.1002/wps.20481>].

Author response: Thanks for this comment. We agree that this is an important issue, and we have now added a section about this in the Intro:

"Pramipexole is a promising candidate drug for anhedonic depression for several different reasons. Drug repurposing, i.e. identifying new use for an already approved drug outside the scope of the original indication 25, can be a powerful and cost-efficient strategy for advancing therapeutic strategies, which has been advocated for in psychiatry 26 27. Our study is a clear example of drug repurposing which has several advantages over the development of a novel drug. Firstly, the risk of failure is lower since i) the safety profile of the repurposed drug has previously been established and ii) the time to implementation is generally shorter. Also, drug repurposing is considered cost-efficient if generic medications are available which is the case of pramipexole. "

2. Perspectives in research on drug repurposing in psychiatry should be briefly discussed in view of recent, relevant evidence in the field. In this regard, the Authors should refer to a recent umbrella review on the topic by Bartoli et al. [2021, <https://doi.org/10.1016/j.jpsychires.2021.09.018>]. The Authors should consider that for pramipexole in depression (as for many other repurposed drugs) currently available evidence is limited – with most meta-analyses being underpowered and only few of them having significant results – and as of today it does not allow making reliable recommendations

for the use of these strategies in clinical practice.

Author response: Thanks for this comment. A clarification regarding the currently limited evidence has been added to the Introduction section. We now cite both the paper by Bartoli et al and the paper by Fava.

Reviewer: 3

Dr. Fredrik Hieronymus, University of Gothenburg

Comments to the Author:

An interesting and highly clinically relevant study which appears to me well designed. Will be looking forward to hearing of the results.

Author response: Thank you for this comment.

VERSION 2 – REVIEW

REVIEWER	Sundberg, Isak Uppsala University
REVIEW RETURNED	17-Oct-2023

GENERAL COMMENTS	No further comments.
----------------------

REVIEWER	Cavaleri, Daniele University of Milan-Bicocca, Department of Medicine and Surgery
REVIEW RETURNED	17-Oct-2023

GENERAL COMMENTS	No further comments.
----------------------